# Occurrence, diversity and distribution of *Trypanosoma* infections in cattle around the Akagera National Park, Rwanda

**Richard Gashururu S.** [1,2,3]*, **Ndichu Maingi**[2], **Samuel M. Githigia**[2], **Methode N. Gasana**[4], **Peter O. Odhiambo**[3], **Dennis O. Getange**[3,5], **Richard Habimana**[1,6], **Giuliano Cecchi**[7], **Weining Zhao**[7], **James Gashumba**[8], **Joel L. Bargul**[3,5], **Daniel K. Masiga**[3]

**1** School of Veterinary Medicine, University of Rwanda, Nyagatare, Rwanda, **2** Faculty of Veterinary Medicine, University of Nairobi, Nairobi, Kenya, **3** International Centre of Insect Physiology and Ecology (*icipe*), Nairobi, Kenya, **4** Rwanda Agriculture and Animal resources Board, Kigali, Rwanda, **5** Department of Biochemistry, Jomo Kenyatta University of Agriculture and Technology, Nairobi, Kenya, **6** Rwanda Food and Drugs Authority, Kigali, Rwanda, **7** Food and Agriculture Organization of the United Nations (FAO), Animal Production and Health Division, Rome, Italy, **8** Rwanda Polytechnic, Kigali, Rwanda

* gasirich@yahoo.fr, r.gashururu@ur.ac.rw

**Data Availability Statement:** All relevant data are within the manuscript and its Supporting Information files.

## Abstract

### Background

African Trypanosomiases threaten the life of both humans and animals. Trypanosomes are transmitted by tsetse and other biting flies. In Rwanda, the African Animal Trypanosomiasis (AAT) endemic area is mainly around the tsetse-infested Akagera National Park (NP). The study aimed to identify *Trypanosoma* species circulating in cattle, their genetic diversity and distribution around the Akagera NP.

### Methodology

A cross-sectional study was carried out in four districts, where 1,037 cattle blood samples were collected. The presence of trypanosomes was determined by microscopy, immunological rapid test VerY Diag and PCR coupled with High-Resolution Melt (HRM) analysis. A parametric test (ANOVA) was used to compare the mean Packed cell Volume (PCV) and trypanosomes occurrence. The Cohen Kappa test was used to compare the level of agreement between the diagnostic methods.

### Findings

The overall prevalence of trypanosome infections was 5.6%, 7.1% and 18.7% by thin smear, Buffy coat technique and PCR/HRM respectively. Microscopy showed a low sensitivity while a low specificity was shown by the rapid test (VerY Diag). *Trypanosoma (T.) congolense* was found at a prevalence of 10.7%, *T. vivax* 5.2%, *T. brucei brucei* 2% and *T. evansi* 0.7% by PCR/HRM. This is the first report of *T.evansi* in cattle in Rwanda. The non-pathogenic *T. theileri* was also detected. Lower trypanosome infections were observed in Ankole x Friesian breeds than indigenous Ankole. No human-infective *T. brucei rhodesiense*

**Funding:** This work received the financial support from Rwanda Dairy Development Project (RDDP); project ID: 2000001195, funded by the International Fund for Agricultural Development (IFAD), through the Rwanda Agriculture and Animal Resources Board (https://www.ifad.org/en/web/operations/-/project/2000001195), to RGS. The support was in the framework of a Memorandum of Understanding (MoU) between the University of Rwanda and the Ministry of Agriculture and Animal Resources (MINAGRI). We are grateful for the International Centre of Insect Physiology and Ecology (icipe), through a Dissertation Research Internship Programme (DRIP) (Postgraduate Training | icipe - International Centre of Insect Physiology and Ecology), Fellowship to RGS. This fellowship also received financial support from the German Ministry for Economic Cooperation and Development (BMZ) through the Deutsche Gesellschaft für Internationale Zusammenarbeit (GIZ) ICTDL Project Contract Number 81235250 and Project Number 18.7860.2-001.00. The funders had no role in study design, data collection and analysis, decision to publish, or preparation of the manuscript.

**Competing interests:** The authors have declared that no competing interests exist.

was detected. There was no significant difference between the mean PCV of infected and non-infected animals (p>0.162).

## Conclusions

Our study sheds light on the species of animal infective trypanosomes around the Akagera NP, including both pathogenic and non-pathogenic trypanosomes. The PCV estimation is not always an indication of trypanosome infection and the mechanical transmission should not be overlooked. The study confirms that the area around the Akagera NP is affected by AAT, and should, therefore, be targeted by the control activities. AAT impact assessment on cattle production and information on the use of trypanocides are needed to help policy-makers prioritise target areas and optimize intervention strategies. Ultimately, these studies will allow Rwanda to advance in the Progressive Control Pathway (PCP) to reduce or eliminate the burden of AAT.

## Author summary

African Trypanosomiasis is a major neglected tropical disease associated with rural areas in low resource settings. The socio-economic and health impact of the disease on humans and livestock is often found at the edge of tsetse-infested protected wildlife areas. Trypanosomiasis is reported around Akagera region of Rwanda at the border with Tanzania, where it is not well documented. This work was the first large-scale study to map *Trypanosoma* occurrence in cattle around the tsetse–infested Akagera National Park. The study determined the genetic diversity and distribution of trypanosomes circulating in cattle blood by using microscopy, immunological rapid tests and molecular techniques. We found animal pathogenic trypanosomes (i.e. *T. brucei brucei*, *T. congolense* savannah, *T. evansi* and *T. vivax*) and the non-pathogenic *T. theileri*. We did not find human-infective *T. b. rhodesiense* causing sleeping sickness. This new knowledge contributes to a better understanding of the epidemiology of animal Trypanosomiasis and it will inform the setting of adequate and more focused control of the disease in the area. The findings are expected to promote the progressive reduction or the elimination of the Animal African Trypanosomiasis burden in the area and inform the process for validation of *rhodesiense* Human African Trypanosomiasis (rHAT) elimination.

## Introduction

African Trypanosomiases constitute a group of vector-borne parasites causing African Animal Trypanosomiasis (AAT) [1] and Human African Trypanosomiasis (HAT) or "sleeping sickness" [2]. The disease is transmitted cyclically by tsetse flies [3,4] and, in the case of some animal-infective trypanosomes, in particular *T. vivax*, mechanically by biting flies such as Tabanids and Stomoxys [5,6]. All tsetse species (Genus: *Glossina*) can transmit the disease, but the savannah species (morsitans group) are the most effective vectors of trypanosomes to livestock [7]. The savannah species cause the utmost threat because they reside in places where animals are usually reared [8,9]. AAT occurs in poor and vulnerable settings of Africa, where it is still overlooked by funders and even the endemic countries themselves. In comparison to other diseases, AAT is often neglected by veterinary authorities because it mainly affects poor

livestock keepers and frequently shows a chronic presentation [10]. Sleeping sickness threatens more than 50 million people in Africa and is characterised as a neglected tropical disease [11–13]. The tsetse and trypanosomiasis challenge to livestock and humans is often linked to the wilderness [14]. This situation increases the exposure to tsetse bite in farming zones around the infested protected area [15].

In Rwanda, the Akagera National Park (NP) and its surroundings are a refuge for tsetse flies [16] and a source of trypanosomal infections [17,18]. In terms of epidemiology, this is an area in which the tsetse fly challenge to livestock is mainly found at the edge of a tsetse-infested park and AAT impact on livestock is highest along the wildlife-livestock interface. The park shelters wild animals, such as buffaloes and warthogs [19,20], which are the natural reservoirs of trypanosomes infections. Around the park there are many cattle farms and communities of farmers [21] which are at risk of diseases occurrence, originating from wild animals and/ or shared between humans, livestock and wild animals [22]. Tsetse control is being implemented inside and outside the park. The park management installs Tsetse traps and target screens, while the ministry of agriculture raises farmers' awareness through campaigns for good pasture management and clearing of the unwanted bush [23]. Tsetse–transmitted trypanosomes have been prevalent and reported around Akagera NP. Farmers rearing livestock in this area are aware of AAT [17,18], and even AAT was detected in cattle and tsetse flies vectors [24]. Sporadic cases of *T. b. rhodesiense* sleeping sickness were last diagnosed in Rwanda around the year 1990 [25]. Presently, there is an adequate surveillance system for HAT, and no case of *T. b. rhodesiense* sleeping sickness has been reported in the country for over the last 20 years [13,26,27], though the area can still be considered at marginal risk. In 2018, a team of the African Union—Pan-African Tsetse and Trypanosomiasis Eradication Campaign (PATTEC) visited Rwanda for monitoring and evaluation to review the development of programmes and strategies aimed at controlling tsetse and Trypanosomiasis in the country. One of the recommendations made was to gather accurate baseline information on tsetse and Trypanosomiasis [28].

The interface between the livestock and the wild reservoirs plays a significant role in the AAT epidemiology in infested areas [15]. In neighbouring Uganda, It has been shown that cattle present the risk of transmission and contribute to the spread of *T. b. rhodesiense* HAT [29–31].

Several diagnostic tests are used to detect trypanosomes, i.e. parasitological [32], Immunological [33,34] and molecular techniques [35]. Although different diagnostic tests for trypanosomes differ in sensitivity, each technique presents its advantages and drawbacks [36,33,37,38]. However, their respective results may serve diversely according to the purpose. The parasitological methods are already in use and the immunological rapid test (VerY Diag) is commercialised in the area. PCR has not been systematically used for the detection of trypanosomes in Rwanda. The study was conducted concurrently with the tsetse survey as a piece of complementary information on the epidemiological parameters of vector distribution and disease risk. This study determined the disease status and identified the trypanosome species circulating in cattle farmed around ANP. The generated findings are crucial for designing evidence-based strategies for AAT control in the area.

## Methods

### Ethics statement

The research ethical permission was approved by the ethics committees of the Faculty of Veterinary Medicine—University of Nairobi (REF: FVM BAUEC/2019/246) and the College of Agriculture and Veterinary Medicine–University of Rwanda (REF: 030/19/DRI).

## Study area

The study area is located in the Eastern Province of Rwanda, near the border with Tanzania in the East and Uganda in the North (Fig 1). The study area was selected for its proximity to the park and adjacent protected game reserves in Tanzania. The three districts where the Akagera NP is located, i.e. Kayonza, Gatsibo and Nyagatare, were the main target, but some data were also collected from the Kirehe District further south. Because of the availability of grazing land [39], the area is dedicated to livestock production with 40% of the national cattle population. Cattle are the dominant domestic animals kept in the area, the indigenous Ankole breed is predominant in the districts of Kayonza and Gatsibo, while crossbreed Ankole x Friesians are the main breeds kept in Nyagatare. Genetic improvement is increasing through cross-breeding with bulls and artificial insemination. Other small livestock species such as goats are also found. The husbandry system is open grazing on individual farms often fenced by *Euphorbia tirucalli* or on the open lands along the park boundary. Cattle farms are concentrated along the interface area with Akagera NP and are consequently exposed to tsetse fly challenge and Trypanosomiasis. The *Glossina* species infesting the area throughout the year are *G. pallidipes* and *G. morsitans centralis*, with an increased abundance during the rainy season [16].

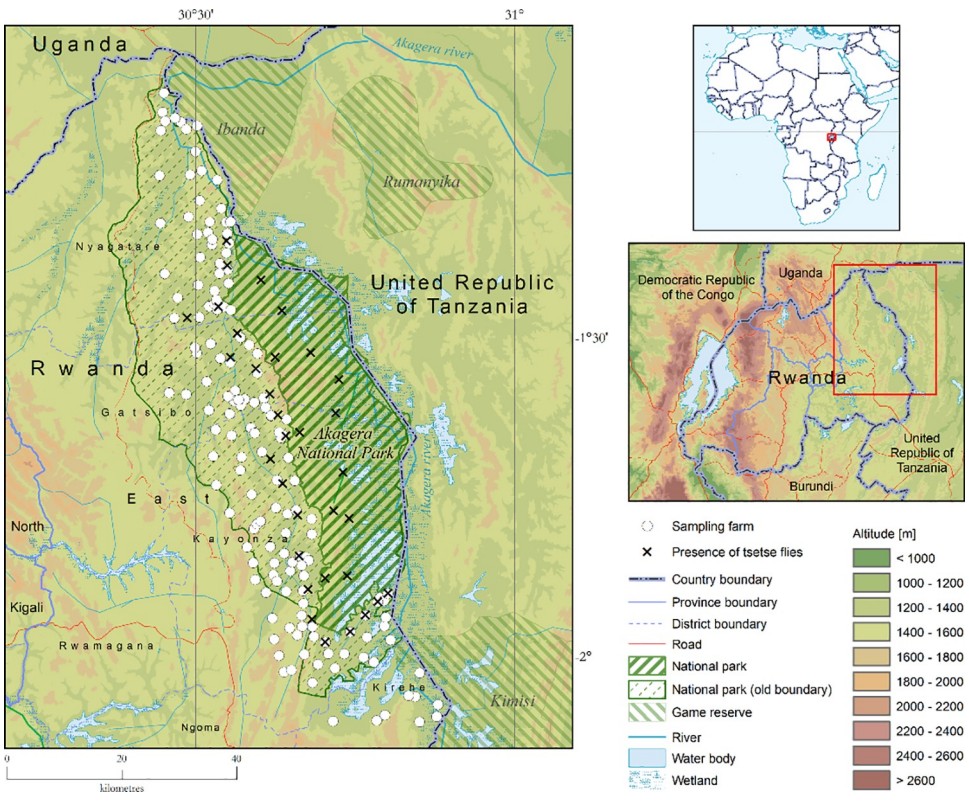

**Fig 1. Study area.** This map was made using the data from the following GIS source files: (1) Digital Elevation Model–STRM https://www.usgs.gov/centers/eros/data-tools; (2) Protected Areas–WDPA https://www.protectedplanet.net/country/RWA (3) Global Administrative Unit Layers (GAUL) https://data.apps.fao.org/map/catalog/srv/eng/catalog.search;jsessionid=B7AF7A215B16770A1A67C65D97FF21CA?node=srv#/metadata/9c35ba10-5649-41c8-bdfc-eb78e9e65654 (4) Inland water bodies in Africa https://data.apps.fao.org/map/catalog/srv/eng/catalog.search;jsessionid=B7AF7A215B16770A1A67C65D97FF21CA?node=srv#/metadata/bd8def30-88fd-11da-a88f-000d939bc5d8 (5) Rivers of Africa https://data.apps.fao.org/map/catalog/srv/eng/catalog.search;jsessionid=B7AF7A215B16770A1A67C65D97FF21CA?node=srv#/metadata/b891ca64-4cd4-4efd-a7ca-b386e98d52e8.

The eastern lowlands of Rwanda have an altitude that ranges between 1 100 m and 1 500 m. The average rainfall is around 1 000 mm per annum but is often irregular with recurrent dry spells. The temperature varies between 19˚C to 29˚C [40]. The vegetation cover includes grassland, woodland and, in proximity to the Akagera NP, bushland [16].

## Study design and sample size determination

A cross-sectional study was undertaken between March and July 2019, using a stratified multistage random sampling method. The sample size was calculated using the formula for indefinite population according to Thrusfield [41]. It was determined based on the baseline-estimated prevalence of 50%, with an absolute desired precision of 5% at the confidence interval of 95%. There was no reliable data on farm records, farming households and cattle population in the area. To increase the chance of sampling many animals in the whole study area, the same calculations were made for each of the three main target districts (stage 1): Kayonza (n = 384), Nyagatare (n = 384) and Gatsibo (n = 236). The target sample size in Gatsibo District was not reached because we could not get consent from some farmers. We purposively selected sectors (stage 2) based on their proximity to Akagera Park, its connections and adjacent game reserves in Tanzania. In that way, one sector of the fourth District (Kirehe) was later included (n = 33), thus making a total sample size of 1037 cattle. These comprised 521 (50.24%) Ankole, 514 (49.56%) Ankole x Friesian and 2 (0.19%) pure Friesian cattle. There were more females (n = 946) than males (n = 91) and the majority were above 2 years of age (n = 876). In total,12 sectors were included in the study (6 in Kayonza, 3 in Nyagatare, 2 in Gatsibo and 1 in Kirehe Districts), and therefore considered as the strata.

Information such as location, cattle population, herd size, communal watering and gathering points was obtained from the local Veterinary Services. At the farm level, individual Ankole and Ankole x Friesian cattle of above 6 months of age and both sexes were randomly selected. Calves aged less than 6 months were excluded because they are less likely to be exposed to AAT, considering the adopted local management system. Farmers do not take the young calves to risky areas for grazing. Cattle below 2 years of age were considered as young and those above 2 years as adults according to cattle owners' information. All sampling sites were georeferenced.

## Blood collection

Blood samples from 1,037 cattle were collected from four districts (Fig 2). Before blood collection, informed oral consent was obtained from farmers. About four (4) mL of blood was collected from the coccygeal vein of each animal using sterile needles and ethylene diamine tetra acetic acid (EDTA) vacutainer tubes (Vacutest Kima, Italy) and each tube was given a unique identifier code. Thin blood smears were immediately prepared on-site, and the remaining blood was transported in cool boxes containing ice blocks to the laboratory of the Rwanda agriculture board. At the same laboratory, the buffy coat technique (BCT) was carried out and the specimens for PCR were prepared. An aliquot of 500 μl of blood was transferred into cryovials and mixed with Lysis, Storage and Transportation (LST) buffer [42] at a ratio of 1:1, and then transported to the International Centre of Insect Physiology and Ecology (icipe), Nairobi-Kenya for molecular analysis.

## Parasitological examination—thin smear and buffy coat technique

The smears that were prepared, fixed in methanol and stored after blood collection were Giemsa stained and examined for trypanosomes by light microscope at a magnification of 100 (Opta-Tech Ltd, Poland). For the buffy coat technique (BCT), haematocrit tubes were sealed

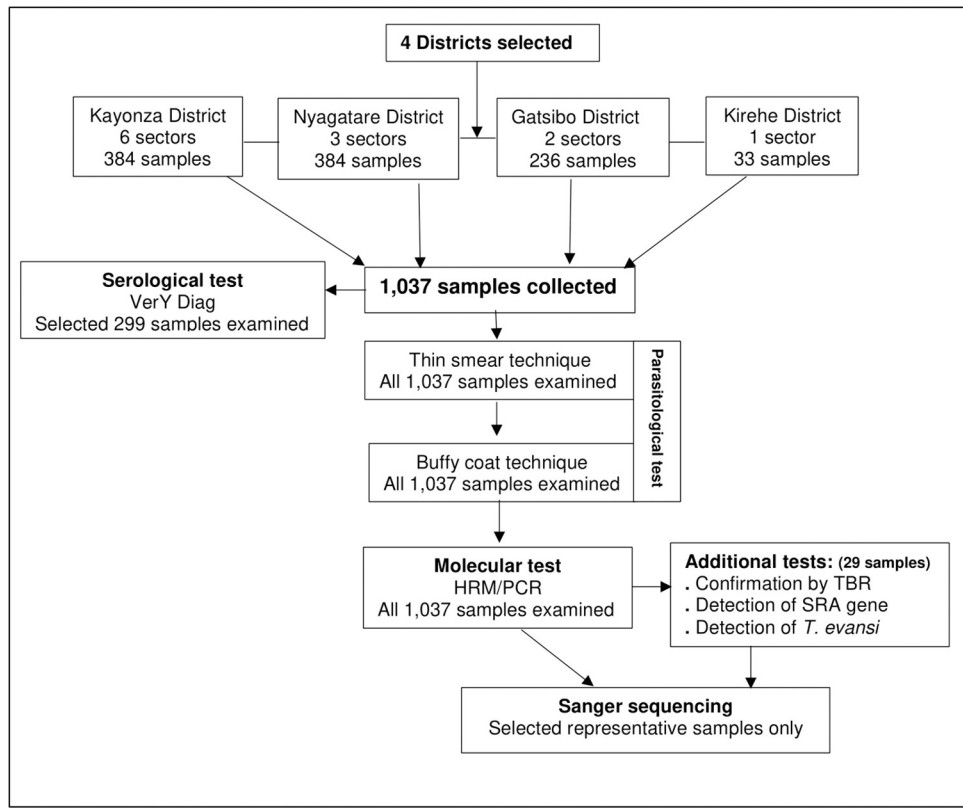

**Fig 2. Schematic study flow.**

with Cristaseal (Hawksley, UK) and centrifuged at 13,000 revolutions per minute (rpm) for 5 min. Hawksley reader was used to determine the packed cell volumes (PCV) levels of each animal and the Buffy coat smears made. The PCV was measured to check correlations of anaemia with trypanosome infections. The Buffy coat smear slides were prepared and observed under a light microscope (Opta-Tech Ltd, Poland) for the presence of trypanosomes as described by [32].

## Nucleic acid extraction

Genomic DNA was extracted from 1037 blood samples using two methods due to logistic challenges of early COVID-19 time. The DNA from the first batch of samples was extracted using Bioline Isolate II genomic DNA kit (Meridian Life Science company, Memphis, TN, USA) as described by the manufacturer's instructions. Genomic DNA from another batch of samples was extracted by the Non-enzymatic salting-out method as described by Saguna [43]. Thereafter, the purity and quantity of the DNA were measured by Eppendorf BioSpectrometer (Enfield, CT, USA); while the quality was measured by 1.5% agarose gel electrophoresis and visualised under UV light.

## Molecular detection of trypanosomes

PCR combined with High-Resolution Melting (HRM) analysis using 18S generic primers (Table 1) were carried out in a volume of 10 μL reaction consisting of 6 μL of nuclease-free water, 2 μL of 5X Hot FIREPol EvaGreen HRM Mix (Solis BioDyne, Tartu, Estonia), 0.5 μL of

**Table 1. Primers used in this study.**

| Primer name | Target gene / specificity | Sequence (5' to 3') | product size / range (bp) | Reference |
|---|---|---|---|---|
| 18S – 3F<br>18S- 4R | 18S rRNA<br>Trypanosomes | GACCRTTGTAGTCCACACTG<br>CCCCCTGAGACTGTAACCTC | 199–241 | [45] |
| ILO 7957<br>ILO 8091 | RoTat1.2 VSG<br>*T.evansi* subtype A | GCC ACC ACG GCG AAA GAC<br>TAA TCA GTG TGG TGT GC | 530 | [46] |
| Eva B1<br>EVAB2 | *T.evansi* subtype B | CACAGTCCGAGAGATAGAG<br>CTGTACTCTACATCTACCTC | 436 | [47] |
| TBR 1<br>TBR 2 | *T. brucei* | CGA ATG AAT ATT AAA CAA TGC GCA GT<br>AGA ACC ATT TAT TAG CTT TGT TGC | 177 (repetitive) | [29] |
| B537<br>B538 | SRA gene | CCATGGCCTTTGACGAAGAGCCCG<br>CTCGAGTTTGCTTTTCTGTATTTTTCCC | 743 | [29] |
| SRA A<br>SRA E | SRA gene | GACAACAAGTACCTTGGCGC<br>TACTGTTGTTGTACCGCCGC | 460 | [44] |

each primer at 10 mM concentrations and 1 μL of DNA template. The PCR conditions were as follows: initial enzyme activation at 95˚C for 12 minutes followed by 40 cycles of denaturation at 95˚C for 30 seconds, annealing at 60˚C for 30 sec and elongation at 72˚C for 1 minute. The final elongation step was set at 72˚C for 10 minutes. Separation was done after PCR amplification from 70 $^0$C to 99.9 $^0$C at the rate of 0.5˚C/ sec. Each run comprised of known positive controls and negative control (PCR mix without nucleic acid template) on which the analysis of amplified PCR products and melt profiles were based. The PCR/HRM analysis was performed on a QuantStudio 3 system (Applied biosystems by Thermo Fisher Scientific).

## Detection of *T. evansi* from *Trypanozoon* positive samples

After the initial molecular screening by PCR/HRM, all *Trypanozoon*—positive samples were subsequently analysed to detect *T. evansi*. Subtypes A and B were targeted by PCR in a ProFlex thermocycler (Applied Biosystems by Life technologies). *T. evansi* Subtype A was screened using ILO F/R primers. 10 μL volume reaction containing 3 μL of nuclease-free water, 5 μL of 2X DreamTaq Green Master Mix (Thermo Fisher Scientific), 0.5 μL of each primer at 10 mM concentrations and 1 μL of DNA template. The PCR conditions were as follows: 95˚C for 1 min, 35 cycles of 94˚C for 30 sec, 58˚C for 30 sec, 72˚C for 30 sec, and a final elongation step of 5 min at 72˚C. Eva B1/B2 primers were used to detect *T. evansi* Subtype B in a 10 μL volume reaction containing 3 μL of nuclease-free water, 5 μL of 2X DreamTaq Green Master Mix (Thermo Fisher Scientific), 0.5 μL of each primer at 10 mM concentrations and 1 μL of DNA template. The PCR conditions were 95˚C for 1 min, 35 cycles of 94˚C for 30 sec, 60˚C for 30 sec, and 72˚C for 1 min, with a final elongation step of 10 min at 72˚C.

## Tests for human infective *T. brucei rhodesiense*

To assess the presence of human-infective trypanosomes in the cattle blood, all the samples positive for *Trypanozoon* were further subject to PCR with TBR primers (Table 1) to confirm their identity. The TBR positive samples were subsequently tested by amplifying the Serum Resistance-Associated (SRA) gene using B537/537 [29] and SRA A/E primers [44]. SRA gene is specific for *T. brucei rhodesiense* and confers resistance to survive in human serum. The PCRs were performed in a ProFlex thermocycler (Applied Biosystems by Life technologies) in a 10 μL volume reaction containing 3 μL of nuclease-free water, 5 μL of 2X DreamTaq Green Master Mix (Thermo Fisher Scientific), 0.5 μL of each primer at 10 mM concentrations and 1 μL of DNA template. The PCR conditions for TBR were as follows: 95˚C for 3 min, 40 cycles of 94˚C for 30 sec, 55˚C for 30 sec, and 72˚C for 1 min, with a final elongation step of 10 min

at 72˚C. The touchdown PCR was used to amplify the SRA gene with B537/538. The conditions were 95˚C for 3 min, followed by 10 cycles of 94˚C for 20 seconds, 55˚C for 30 seconds and 72˚C for 1 min, followed 25 cycles of 94˚C for 20 seconds, annealing at 63.8˚C for 30 seconds and extension at 72˚C for 1 min per cycle. The final extension was set at 72˚C for 7 min. The PCR conditions for SRA A/E were as follows: 95˚C for 3 min, 40 cycles of 95˚C for 1 min, 68˚C for 1 min and 72˚C for 1 min, with a final elongation step of 10 min at 72˚C.

## Sequence analysis

PCR amplicons for the positive samples were run in 2% ethidium stained agarose gel electrophoresis, and the target ones with a correct single band were purified by Exo1-rSAP (New England BioLabs, inc. MA, US) as instructed in the guideline. The products with more than one band were excised and then purified by a QIAquick PCR purification kit (Qiagen, Germany). The purified amplicons were sequenced unidirectionally at Macrogen Inc. (Holland). The resultant sequence chromatograms were processed using Geneious prime (version 20.2.2) (Biomatters, New Zealand). To identify sequence homology, sequence nucleotides were compared by BLAST with sequences deposited in the NCBI GenBank database. Maximum likelihood phylogenies were inferred using PhyML version 3.0. An Akaike information criterion for automatic selection for an appropriate model of evolution was employed during the phylogeny construction. The generated tree was visualized and edited in Figtree 1.4. Pairwise genetic distances were conducted in MEGA software version 7 using the Tajma-Nei model.

## Immunological rapid diagnostic test (VerY Diag)

Among the 1,037 blood samples collected, two hundred ninety-nine (299) were randomly selected and used for a rapid test called VerY Diag [34]. The test was done on fresh blood samples after collection. The number of samples was limited to 299 because of the cost of the test kit ($6 per single test). VerY Diag is a lateral flow rapid field test with immune-chromatography designed to detect *T. congolense* and *T. vivax* species. It was developed using recombinant antigens TcoCB1 (test line Tc, 0.65 mg/mL) and TvGM6 (test line Tv, 1.2 mg/mL) [34]. With the help of a pipette supplied in the kit, a drop of whole blood (20 μL) was deposited into the specimen well of the cassette. Immediately, another drop (40 μL) of the dilution buffer was added to the same well, avoiding to drop any solution in the cassette observation window. Results were read after 10 minutes.

## Statistical analysis

The data was analysed by descriptive statistics in SPSS software (SPSS Inc., IL, USA). A parametric test (ANOVA) was used to compare the Mean PCV and disease prevalence between different areas. The Cohen Kappa test was used to compare the level of agreement between the different diagnostic methods. The significance threshold was fixed at 5% and 95% of confidence.

## Results

### Trypanosomes detected by parasitological and molecular methods

The overall prevalence of trypanosome infections by thin smear was 5.6%, of which *T. congolense* accounted for 3.5% (n = 37/1037*)*, *T. vivax* 1.9% (n = 20/1037), *Trypanozoon* 0.09% (n = 1/1037) and the inconclusively identified trypanosomes 0.28% (n = 3/1037). The Buffy coat technique increased the overall prevalence to 7.1% of which 5.1% (n = 53/1037) were *T. congolense*, 2.4% (n = 25/1037) of *T. vivax*, 0.09% (n = 1/1037) of *Trypanozoon* and 0.86%

**Table 2. Trypanosomal infections by different diagnostic tests.**

| District | Sector | NE | Thin smear | | | | | | Buffy coat technique | | | | | | PCR/HRM | | | | | | |
|---|---|---|---|---|---|---|---|---|---|---|---|---|---|---|---|---|---|---|---|---|---|
| | | | Tc | Tv | Tz | Over. Prev. | U | Mixed | Tc | Tv | Tz | Over. Prev. | U | Mixed | Tc | Tv | Tbb | Te | Over. Prev. | T. th | Mixed |
| Kayonza | Ndego | 70 | 5 | 2 | 0 | (10%) | 0 | 0 | 7 | 2 | 0 | (12.8%) | 0 | 0 | 10 | 6 | 0 | 2 | (25.7%) | 0 | 1(Tv+T. th) |
| | Kabale | 6 | 0 | 1 | 0 | (16.6%) | 0 | 0 | 0 | 1 | 0 | (16.6%) | 0 | 0 | 0 | 1 | 0 | 0 | (16.6%) | 0 | - |
| | Rwinkwavu | 10 | 0 | 1 | 0 | (10%) | 0 | 0 | 0 | 1 | 0 | (10%) | 0 | 0 | 0 | 4 | 0 | 0 | (40%) | 0 | - |
| | Mwiri | 80 | 1 | 1 | 0 | (2.5%) | 2 | 0 | 1 | 1 | 0 | (2.5%) | 3 | 0 | 3 | 7 | 0 | 0 | (12.5%) | 16 (20%) | - |
| | Gahini | 89 | 3 | 0 | 0 | (3.3%) | 0 | 0 | 5 | 1 | 0 | (6.7%) | 1 | 0 | 8 | 2 | 0 | 1 | (12.3%) | 5 (5.6%) | 1 (Tc+Te) |
| | Murundi | 129 | 6 | 1 | 0 | (5.4%) | 0 | 0 | 16 | 1 | 0 | (13.1%) | 0 | 0 | 42 | 8 | 1 | 1 | (40.3%) | 12 (9.3%) | 1(Tc+Tv) |
| | *subtotal* | *384* | *15* | *6* | *0* | *(5.4%)* | *0* | *0* | *29* | *7* | *0* | *(9.3%)* | *4* | *0* | *63* | *28* | *1* | *4* | *(25%)* | *33 (8.6%)* | |
| Gatsibo | Rwimbogo | 190 | 1 | 3 | 0 | (2.1%) | 0 | 0 | 2 | 6 | 0 | (4.2%) | 0 | 0 | 3 | 12 | 0 | 0 | (7.9%) | 18 (9.4) | 1(Tv+T. th) |
| | Kabarore | 46 | 0 | 0 | 0 | (0%) | 1 | 0 | 0 | 0 | 0 | (0%) | 3 | 0 | 0 | 0 | 2 | 0 | (4.3%) | 19 (41.3%) | - |
| | *Subtotal* | *236* | *1* | *3* | *0* | *(1.7%)* | *1* | *0* | *2* | *6* | *0* | *(3.4%)* | *3* | *0* | *3* | *12* | *2* | *0* | *(7.2%)* | *37 (15.6%)* | |
| Nyagatare | Karangazi | 174 | 18 | 9 | 0 | (15.5%) | 1 | 0 | 17 | 10 | 0 | (15.5%) | 1 | 0 | 37 | 13 | 8 | 4 | (35.6%) | 4 (2.3%) | 4 (Tb+Tc) |
| | Rwimiyaga | 186 | 3 | 2 | 1 | (3.2%) | 1 | 0 | 5 | 2 | 1 | (4.3%) | 1 | 0 | 7 | 1 | 5 | 0 | (7%) | 5 (2.7%) | - |
| | Matimba | 24 | 0 | 0 | 0 | (0%) | 0 | 0 | 0 | 0 | 0 | (0%) | 0 | 0 | 1 | 0 | 4 | 0 | (20.8%) | 0 | - |
| | *Subtotal* | *384* | *21* | *11* | *1* | *(8.6%)* | *2* | *0* | *22* | *12* | *1* | *(9.1%)* | *2* | *0* | *45* | *14* | *17* | *4* | *(20.8%)* | *9 (2.3%)* | |
| Kirehe | Mpanga | 33 | 0 | 0 | 0 | (0%) | 0 | 0 | 0 | 0 | 0 | (0%) | 1 | 0 | 0 | 0 | 1 | 0 | (3%) | 4 (12.1%) | - |
| | *Subtotal* | *33* | | | | *(0%)* | | *0* | | | | *(0%)* | | *0* | | | *1* | *0* | *(3%)* | *4 (12.1%)* | |
| | **Total** | **1037** | **37** | **20** | **1** | **(5.6%)** | **3** | **0** | **53** | **25** | **1** | **(7.1%)** | **9** | **0** | **111** | **54** | **21** | **8** | **(18.7%)** | **83(8%)** | **8 (0.7%)** |

NE = Number of animals examined; Tc = *Trypanosoma congolense*; Tv = *Trypanosoma vivax*; Tz = Trypanozoon; U = unidentified; *T. brucei brucei* = *Trypanosoma brucei brucei*, Te = *Trypanosoma evansi*, *T.th* = *Trypanosoma theileri*, *Over. Prev.* = Overall prevalence. The overall prevalence does not include infection with the non-pathogenic *T. theileri*. Mixed infections are included in overall prevalence as single infections

Non-pathogenic trypanosomes were mostly found in areas of Gatsibo 15.6% and Kirehe (12.1%) districts, but also some parts of Kayonza (8.6%).

(n = 9/1037) of inconclusive trypanosomes. All the inconclusive trypanosomes were suspected to be non-pathogenic, and they were later identified as *T. theileri* by PCR and by sequencing. No mixed infections were detected by microscopy.

The overall prevalence of pathogenic trypanosomes detected by PCR/HRM was 18.7% (n = 194/1,037). Of these, *T. congolense* represented 10.7% (n = 111/1,037), *T. vivax* 5.2% (n = 54/1,037), *Trypanozoon* 2.8% (n = 29/1,037). *Trypanozoon*—positive samples tested by PCR/HRM were subjected to specific primers for *T. brucei brucei* and *T. evansi*, which later gave *T. brucei brucei* 2% (n = 21/1,037) and *T. evansi* 0.7% (n = 8/1,037) (Table 2). The 8 samples positive to *T. evansi* were detected by ILO primer, and are therefore *T. evansi* sub-type A. All samples were negative to Eva B1/B2 primer targeting sub-type B, meaning no *T. evansi* sub-type B was detected in samples examined.

The non-pathogenic *T. theileri* represented 8% (n = 83/1,037). Among the above infections, eight mixed infections (0.7%; n = 8/1037) were found, comprising four infections of *T. brucei brucei* and *T. congolense*, 2 infections of *T. vivax* and *T. theileri*, one infection of *T. congolense* and *T. vivax*, and one infection of *T. congolense* and *T. evansi*. As per PCR/HRM results, *T. congolense* and *T. vivax* were dispersed throughout the three districts. No *T. vivax* was found in Kirehe and *T. brucei brucei* infections were concentrated in Nyagatare. The non-pathogenic

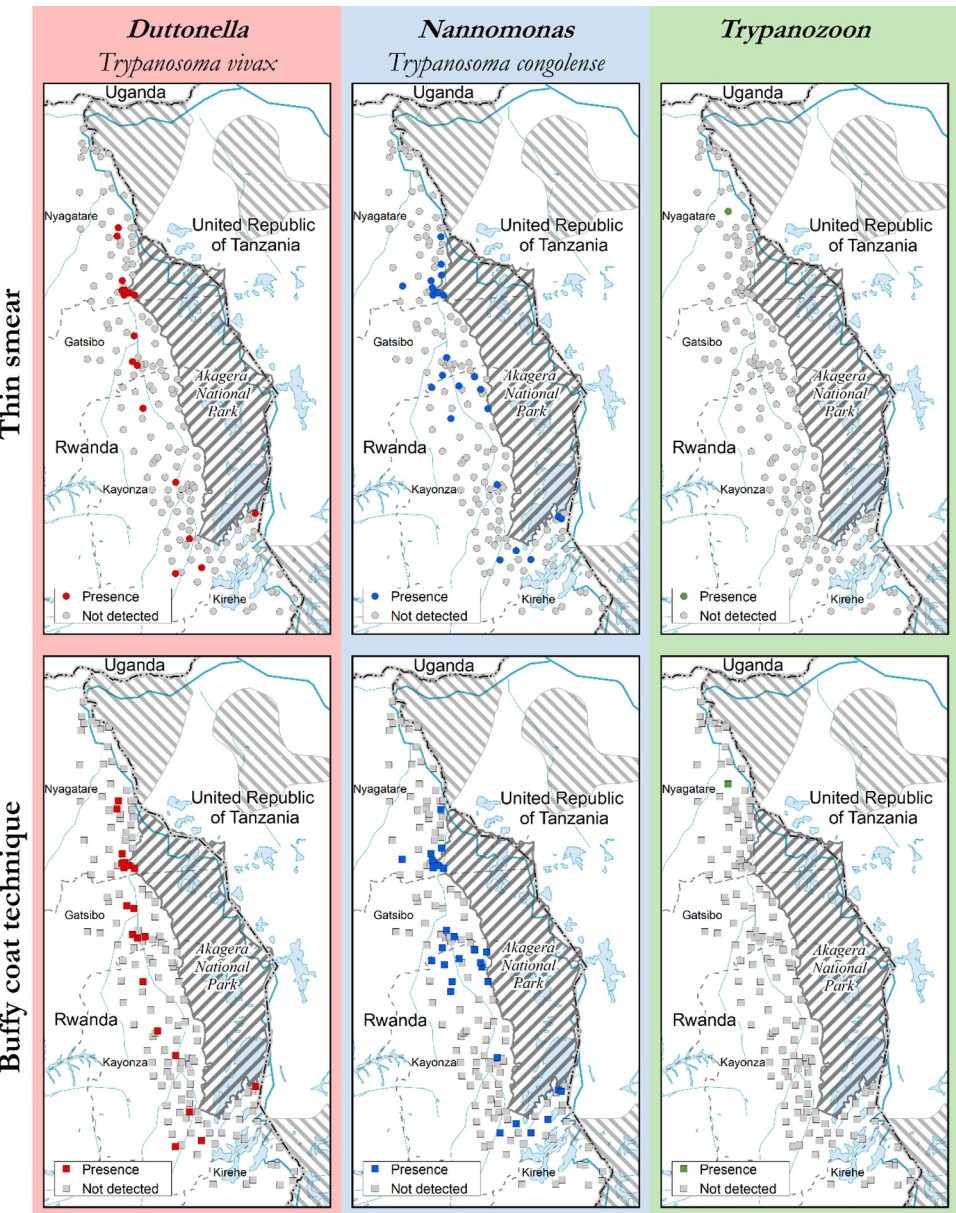

**Fig 3. Distribution of trypanosomes detected by thin smear and Buffy coat technique.** This map was made using the data from the following GIS source files: (1) Protected Areas–WDPA https://www.protectedplanet.net/country/RWA (2) Global Administrative Unit Layers (GAUL) https://data.apps.fao.org/map/catalog/srv/eng/catalog.search;jsessionid=B7AF7A215B16770A1A67C65D97FF21CA?node=srv#/metadata/9c35ba10-5649-41c8-bdfc-eb78e9e65654 (3) Inland water bodies in Africa https://data.apps.fao.org/map/catalog/srv/eng/catalog.search;jsessionid=B7AF7A215B16770A1A67C65D97FF21CA?node=srv#/metadata/bd8def30-88fd-11da-a88f-000d939bc5d8 (4) Rivers of Africa https://data.apps.fao.org/map/catalog/srv/eng/catalog.search;jsessionid=B7AF7A215B16770A1A67C65D97FF21CA?node=srv#/metadata/b891ca64-4cd4-4efd-a7ca-b386e98d52e8.

*T. theileri* was more prevalent in Gatsibo and Kayonza. Figs 3 and 4 show the spatial distribution of trypanosomal infections by diagnostic methods across the study area.

By comparing infection status between breeds, indigenous Ankole cattle were more infected by *T. congolense* (15.7%; 82/521) and *T. vivax* (7.3%; 38/514) than crossbreed Ankole x Friesians (p = 0.000). However, more *Trypanozoon* species were found in crossbreed Ankole x Friesians than indigenous Ankole (Table 3). The occurrence of non-pathogenic infections

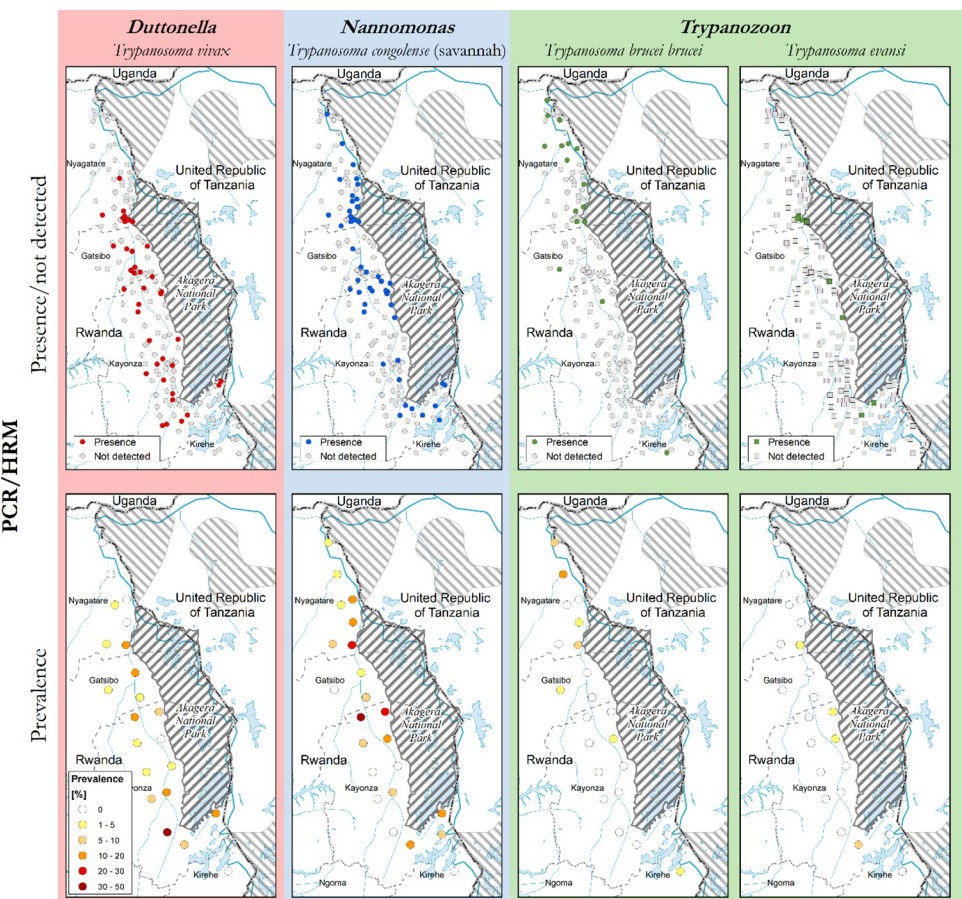

**Fig 4. Distribution of pathogenic trypanosomes detected by PCR/HRM.** This map was made using the data from the following GIS source files: (1) Protected Areas–WDPA https://www.protectedplanet.net/country/RWA (2) Global Administrative Unit Layers (GAUL) https://data.apps.fao.org/map/catalog/srv/eng/catalog.search;jsessionid= B7AF7A215B16770A1A67C65D97FF21CA?node=srv#/metadata/9c35ba10-5649-41c8-bdfc-eb78e9e65654 (3) Inland water bodies in Africa https://data.apps.fao.org/map/catalog/srv/eng/catalog.search;jsessionid=B7AF7A215B16770 A1A67C65D97FF21CA?node=srv#/metadata/bd8def30-88fd-11da-a88f-000d939bc5d8 (4) Rivers of Africa https:// data.apps.fao.org/map/catalog/srv/eng/catalog.search;jsessionid=B7AF7A215B16770A1A67C65D97FF21CA?node= srv#/metadata/b891ca64-4cd4-4efd-a7ca-b386e98d52e8.

between the two breeds was very similar. The number of Friesian cattle was too small to be considered in comparison, therefore not included in this table.

As shown in Fig 5, non-pathogenic trypanosomes were more prevalent in Gatsibo and Kayonza districts than in the north (Nyagatare district) and south (Kirehe district). HRM melt curve profiles and their alignment (Fig 6) were the basis of the identification of trypanosomes.

## Correlation of packed cell volume with Trypanosomal infection

The overall mean PCV of infected animals was 29.5 for PCR and 28.5 for microscopy compared to 30.4 and 30.3 observed in non-infected animals, respectively for PCR and microscopy. This difference did not show any significant effect on the trypanosome infection for PCR (p>0.164) and microscopy results (p>0.212). There was even a negative Pearson correlation between the PCV values of negative and the positive results (r = -0.007). PCV was not determined for 2 animals due to the poor quality of the blood after centrifugation, however, no trypanosomes were detected in the same animals for all the methods used. Lower and higher

**Table 3. Other predictors of infection with different trypanosome species [Positive by HRM-PCR].**

| Predictor | NE | Pathogenic infections | | | | | Non-pathogenic | Mixed infections | | | |
|---|---|---|---|---|---|---|---|---|---|---|---|
| Breed | | *Tc* | *Tv* | *Tbb* | *Te* | *Total* | *T.th* | *Tbb +Tc* | *Tc+Tv* | *Tc+Te* | *Tv+T.th* |
| Ankole | 521 | 82 (15.7%) | 38 (7.3%) | 8 (1.5%) | 4 (0.7%) | 132 (25.3%) | 42 (8%) | 2 (0.38%) | 1 (0.2%) | 1 (0.2%) | 0 |
| Ankolex Friesian | 514 | 29 (5.6%) | 16 (3.1%) | 13 (2.5%) | 4 (0.7%) | 62 (12%) | 41 (7.97%) | 2 (0.38%) | 0 | 0 | 1 (0.2%) |
| Friesian | 2 | 0 | 0 | 0 | 0 | 0 | 0 | 0 | 0 | 0 | 0 |
| Sex | | | | | | | | | | | |
| Female | 946 | 105 (11.1%) | 43 (4.5%) | 20 (2.1%) | 8 (0.8%) | 176 (18.6%) | 78 (8.2%) | 4 (0.4%) | 1 (0.1%) | 1 (0.1%) | 2 (0.2%) |
| Male /Neutered | 91 | 6 (6.5%) | 11 (12.1%) | 1 (1.08%) | 0 | 18 (19.8%) | 5 (5.4%) | 0 | 0 | 0 | 0 |
| Age | | | | | | | | | | | |
| < 2 years [Young] | 161 | 11 (6.8%) | 16 (9.9%) | 1 (0.6%) | 0 | 28 (17.4%) | 10 (6.2%) | 0 | 0 | 0 | 0 |
| >2years [Adults] | 876 | 100 (11.4%) | 38 (4.3%) | 20 (2.2%) | 8 (0.9%) | 166 (18.9%) | 73 (8.3%) | 4 (0.4%) | 1 (0.1%) | 1 (0.1%) | 2 (0.2%) |
| **Overall** | **1037** | **111 (10.7%)** | **54 (5.2%)** | **21 (2%)** | **8 (0.7%)** | **194 (18.7%)** | **83 (8%)** | **4 (0.38%)** | **1 (0.1%)** | **1 (0.1%)** | **1 (0.1%)** |

NE = Number of animals examined; Tc = *Trypanosoma congolense*; Tv = *Trypanosoma vivax*; Tbb = *Trypanosoma brucei brucei*; Te = *Trypanosoma evansi*

T.th = *Trypanosoma theileri*. Non- pathogenic *T. theileri* cases are counted separately. *Mixed infections are counted in single infections.*

PCV values were seen in either group. Looking at individual trypanosome species, the PCV values of infected animals were grouped in thresholds where the PCV of 26% and less was considered anaemic (Table 4). For many positive cases, the PCV was above the threshold of 26%. There was no statistical difference between the two groups of infected animals (p>0.162) and (p>0.212), respectively for PCR and microscopy.

Below the threshold (<26%), there were 86 positive cases against 191 found above the threshold. In the non-infected group, 209 animals had the PCV below the threshold while 551 animals were above 26%. Animals with mixed infections seem to have a lower mean PCV (27.3), followed by the single infections of *T. vivax* (28.9) even though this difference was not statistically significant. For microscopy, 34 positives were below 26 and 55 positives above 26 of PCV.

## Genetic diversity of trypanosome species

The 18S rRNA study sequences of *T. congolense* showed similarity of between 98.71–99.56% with GenBank accession: AJ223563.1(Cattle) and 98.24–99.14% with AJ009146.1 (Goat) and U22315.1—IL1180 (Cattle), all of them being *T. congolense savannah* from Kenya. We found no *T. congolense* forest and *T. congolense* Kilifi subspecies. Our *T. vivax* representatives showed 100.00% similarity with IL3905 GenBank DQ317414 (cattle, Kenya) and 100% with GenBank: KM391821 (cattle, Ethiopia). The two GenBank similarities correspond to the TvL1-G genotype (West Africa & East Africa) of *T.vivax. T. theileri* representatives showed 100.00% similarity with the GenBank accession: KF924256 (cattle, Poland) and AJ009163 (UK). The BLAST results of the representative sequences from *Trypanozoon* gave the highest similarity (100%) to 2 or more species of the group such as *T. brucei brucei* XR002989632 from the UK and *T. evansi* MN446740.1 from China. The amplicons size was not long enough to definitely resolve them. The nucleotide sequences from this study were deposited to the GenBank database under the following accession numbers OK264415, OK264416, OK264417 (*T.brucei*), OK264418, OK264419 (*T.congolense*), OK264420, OK264421 (*T.theileri*), OK264422, OK264423 (*T.vivax*) (Fig 7). Divergence estimates are shown in Table 5.

## Detection of SRA gene in cattle

Out of 29 samples tested positive for *Trypanozoon* by PCR/HRM, 21 were positive to the TBR primer, however negative for *T. evansi*. They were, therefore, classified as *T. brucei brucei*.

None of the 21 TBR positive samples tested positive for the SRA gene by using either SRA A/E or B537/538 primers. This means that no causative agent of *rhodesiense* sleeping sickness was found in the cattle blood analysed.

## Immunological rapid test results (VerY Diag)

Out of 299 animals examined using the VerY Diag rapid test, 296 showed conclusive results while 3 cassettes showed indecisive results. 19 (6.4%) showed antibodies to *T. congolense*, 77 (26%) to *T. vivax* and while 88 (29.7%) animals showed antibodies to both *T. congolense* and *T. vivax*. Trypanosome antibodies were detected more in adult cattle (168/296) than young ones (16/296). 112 samples (37.8%) did not show trypanosome antibodies (Table 6). Fig 8 shows the example for the results of the rapid diagnostic test (VerY Diag), and their distribution is shown in Fig 9. The VerY Diag cassettes technology do not cross-react with trypanozoon species

## Comparison between diagnostic tests

The thin smear and Buffy coat technique detected few trypanosome infections, however, the buffy coat technique showed a higher sensitivity compared to the thin smear. Apart from the increased number of positive cases, PCR/HRM detected mixed infections and considerably more *Trypanozoon* infections. Using the PCR/HRM results as a reference, the sensitivity and specificity of the other diagnostic tests are shown in Table 7.

Thin smear and Buffy coat technique increase the false negatives hence the low sensitivity while VerY Diag resulted in poor specificity. The Cohen Kappa test showed an increased level of agreement between the thin smear and Buffy coat technique (K = 0.807), and a low agreement between thin smear and PCR/HRM (K = 0.310). A moderate agreement coefficient was found between the Buffy coat technique and PCR/HRM (K = 0.424). The immunological rapid test (VerY Diag) only detects antibodies of *T. congolense* and *T. vivax*. The Cohen Kappa test was, therefore, run merely for the two species when compared with other methods. There was a very low agreement of K = 0.037, K = 0.042, K = 0.031 between the thin smear, Buffy coat technique and PCR/HRM respectively.

## Discussion

Our study determined the diversity of trypanosomes circulating in cattle around the Akagera NP in Rwanda and found common pathogenic trypanosomes for cattle (i.e. *T. congolense*, *T. vivax*, *T. brucei* and *T. evansi*) and the non-pathogenic *T. theileri*. The findings on pathogenic trypanosomes are in line with what was reported in the area by Mihok *et al.*, [24]. Our study provides the first report of *T. evansi* in cattle in Rwanda. *T.congolense* was the most abundant species, followed by *T.vivax* and *Trypanozoon* as the least abundant. In particular, *T. congolense* and *T. vivax* seem to be closely associated with the park and its tsetse-infested boundaries [16]. The park shelters the known reservoirs of trypanosomes such as buffaloes and warthogs. Tsetse flies freely feed on both wild animals and livestock. The open grazing management adopted favours the transmission of trypanosomes. The park can be considered presumably as a block of tsetse infestation. However, tsetse-transmitted trypanosomes may be found in tsetse-free areas due to the movement of animals [6] through sales, family migrations or other livestock programmes. The study noted a cluster of *Trypanozoon* species in the north of Akagera NP (Fig 3). The north of the park has a higher concentration of wild animals [19,20] and the same is the case for livestock around the interface. A concomitant study found higher densities of tsetse flies [16] and other biting flies (stomoxys, etc) were observed in the same area during the data collection, which suggests increased transmission in the north. There might be

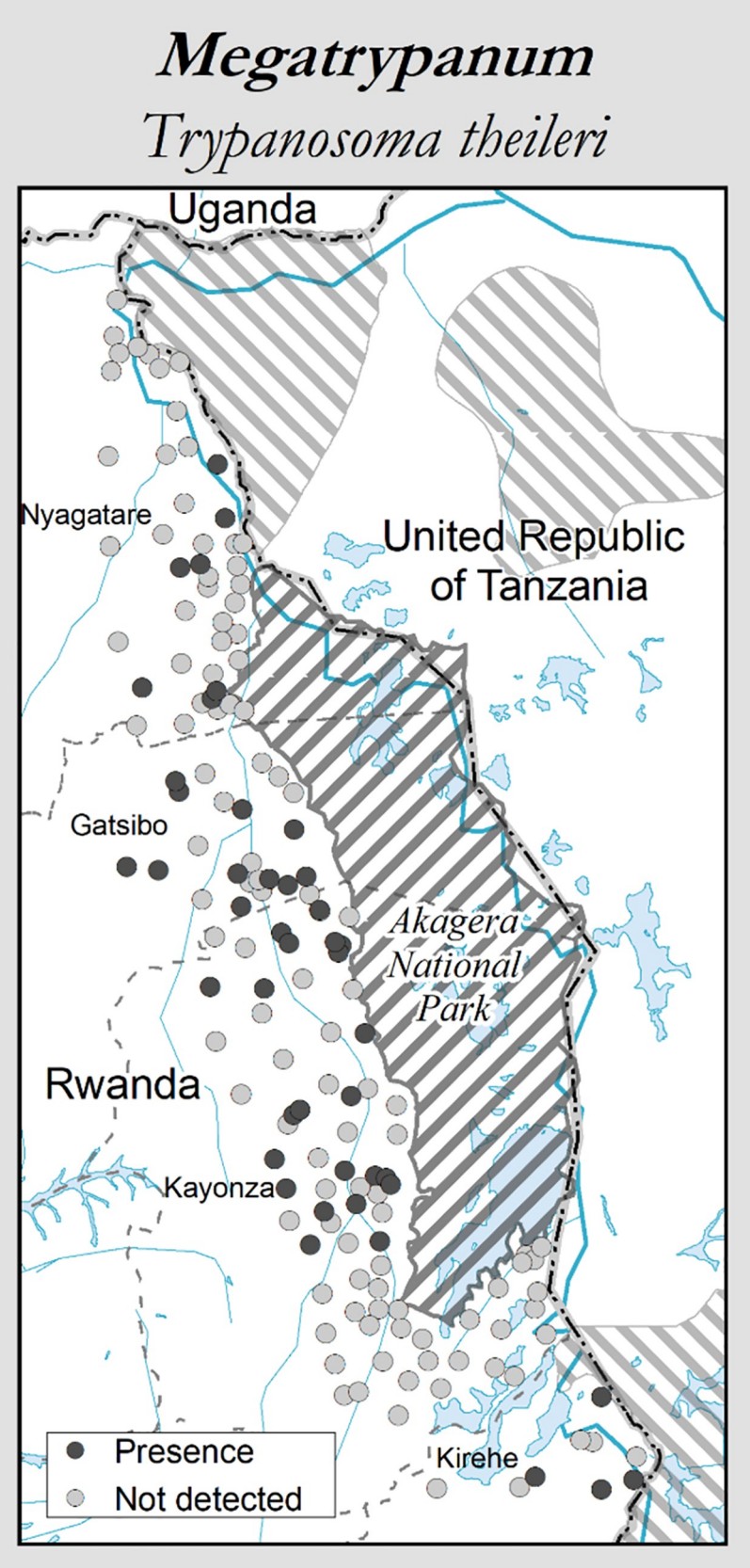

**Fig 5. Distribution of non-pathogenic trypanosomes across the study area.** This map was made using the data from the following GIS source files: (1) Protected Areas–WDPA https://www.protectedplanet.net/country/RWA (2) Global Administrative Unit Layers (GAUL) https://data.apps.fao.org/map/catalog/srv/eng/catalog.search;jsessionid=B7A F7A215B16770A1A67C65D97FF21CA?node=srv#/metadata/9c35ba10-5649-41c8-bdfc-eb78e9e65654 (3) Inland water bodies in Africa https://data.apps.fao.org/map/catalog/srv/eng/catalog.search;jsessionid=B7AF7A215B1 6770A1A67C65D97FF21CA?node=srv#/metadata/bd8def30-88fd-11da-a88f-000d939bc5d8 (4) Rivers of Africa https://data.apps.fao.org/map/catalog/srv/eng/catalog.search;jsessionid=B7AF7A215B16770A1A67C65D97FF21CA? node=srv#/metadata/b891ca64-4cd4-4efd-a7ca-b386e98d52e8.

some preferred hosts harbouring *Trypanozoon* species on which the tsetse flies and other biting flies are feeding. Additionally, this area shares a border with the Ibanda game reserve in Tanzania, which could contribute to the transmission of trypanosomes in the area. There is a need to investigate this situation, although no tsetse flies were collected around this game reserve during the entomological study [16]. *T. evansi* has a wide host range [33] and similar sub-type A was isolated from camels and buffalo in the region [48]. Regionally, this trypanosome species was extensively found and studied in Kenya [48,49].

The non-pathogenic *T. theileri* was found in cattle reared around Akagera NP, as it was also reported previously by Mihok *et al.*, [24]. This benign parasite was found in cattle in Uganda [50] and other African regions [51]. The parasite was mainly detected in Gatsibo and Kayonza districts (Fig 4). *T. theileri* is in the stercorarian group, under the subgenus *Megatrypanum*. The *Trypanosoma theileri* group comprises three species hardly discernible but which are host specific: *T. theileri* for bovine, *T. melophagium* for ovine and *T. cervi* for deer. The host specificity helps, in addition to the sequencing data to identify these parasites [52]. *T. theileri* is transmitted between wild and domestic animals by biting flies (tabanids, *Stomoxys*, etc.). Despite the presence of mechanical vectors in the area, there should be an increased interaction between livestock and some *T. theileri*-specific wild hosts from the park side. Further investigation on this is recommended to evaluate the current pathogenic effect of *T. theileri* group on cattle and /or other livestock species health.

Apart from tsetse flies, biting flies such as tabanids and *Stomoxys* were observed and are good mechanical vectors of some Trypanosome species. The presence of *T. evansi* and *T. theileri* in cattle blood show the importance of other blood-feeding flies in mechanically transmitting the trypanosomes in the area. *T. vivax* can also be transmitted in this way. This indicates a possible role of AAT mechanical transmission in the area, even though the mechanically transmitted trypanosomes cannot survive long outside the host [33]. Biting flies were collected in the area and the related data will be presented elsewhere.

As in many similar studies, the microscopy showed low sensitivity in detecting trypanosomes as compared to molecular techniques [36]. However, the microscopy specificity is still high. The low sensitivity results from subclinical infections expressing low levels of parasitaemia in infected animals. Notwithstanding the consistency, the buffy coat technique confirms its higher sensitivity in that it detected more infections than a thin smear. This is slightly more evident for *T. congolense* infections. Microscopy failed to detect almost all the *Trypanozoon* and *T. theileri*. This could be because of very low parasitemia and affinity for tissues by *T. brucei*, while *T. congolense* and *T. vivax* are mainly intravascular [33]. At microscopy, infections of *T. theileri* were suspected and not identified. However, their identity was later confirmed as *T. theileri* by HRM and sequencing.

Although sensitive, PCR missed some positive cases of trypanosomes that were detected by microscopy. This could be due to the quantity and or quality of parasitic nucleic acid extracted. PCR detects trypanosomes DNA in a sample and is much sensitive compared to other routine diagnostic techniques, of clinical, subclinical and chronic infections for both pathogenic and non-pathogenic trypanosomes [35]. However, the PCR positive cases do not necessarily mean

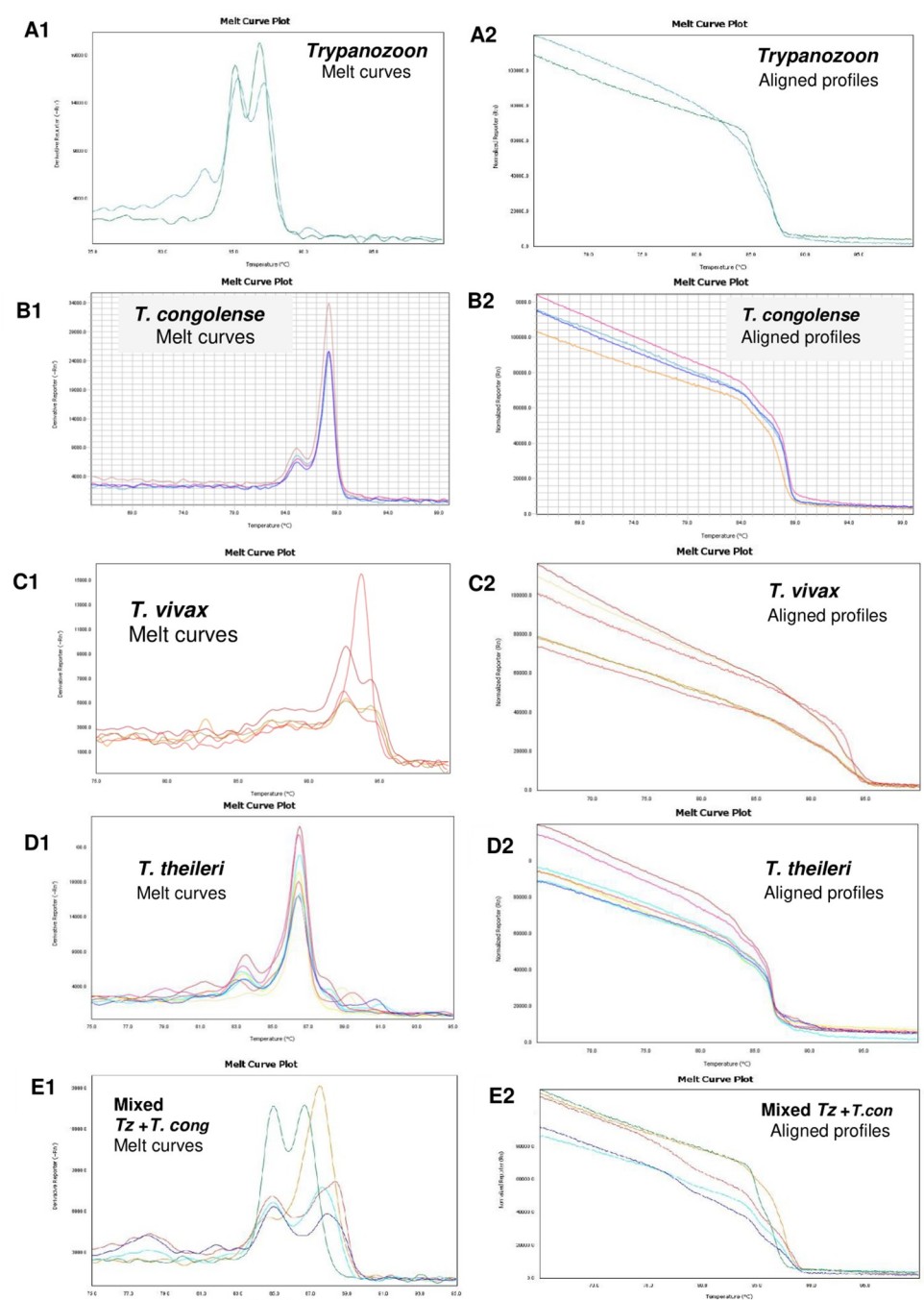

**Fig 6. Melt curve plots and normalised (aligned) HRM profiles of representative positives.** A1-A2: T. brucei s.l. B1-B2: T. congolense C1-C2: T. vivax D1-D2: T. theileri. E1-E2: Mixed infections of Trypanozoon + T. congolense.

the clinical disease. Therefore, the results cannot be directly associated with the disease impact on the production and health status of the cattle in the study area [37,38]. Nevertheless, it should be noted that carrier animals affect the smooth running of control programmes [53]. Additionally, PCR is expensive and less applicable in rural settings where farmers are concerned to know infected animals for treatment. Molecular detection should be recommendable in case the goal is to maximize detection and describe the diversity of trypanosomes. However,

**Table 4. PCV correlation between trypanosome species according to thresholds.**

| Species | Mean PCV | <26% | >26% | Sum |
|---|---|---|---|---|
| **PCR/HRM** | | | | |
| *T. brucei brucei* | 29.7 | 7 | 14 | 21 |
| *T. evansi* | 28.2 | 3 | 5 | 8 |
| *T. congolense* | 29.2 | 39 | 72 | 111 |
| *T. vivax* | 28.9 | 18 | 36 | 54 |
| *T. theileri* | 30.4 | 19 | 64 | 83 |
| Mixed infections | 27.3 | 3 | 5 | - |
| Total | *NA* | 86 | 191 | 277 |
| P value | | 0.162 | | |
| **Microscopy (Buffy coat technique)** | | | | |
| *T. congolense* | 28.4 | 22 | 31 | 53 |
| *T. vivax* | 27.2 | 11 | 14 | 25 |
| Trypanozoon | 31 | 0 | 1 | 1 |
| Unidentified | 32.1 | 1 | 9 | 10 |
| Mixed infections | - | 0 | 0 | |
| Total | NA | 34 | 55 | 89 |
| P value | | 0.212 | | |

NA = Not applicable, mixed infections were counted in single infections

if the goal is to find animals for treatment and to minimise diagnostic cost, the buffy coat technique could be an option.

VerY Diag detects circulating antibodies for *T. congolense* and *T. vivax* and does not discriminate between active, recent and past infections following treatment. The VerY Diag detects more *T. vivax* than *T. congolense* for it has a sensitivity of 92.0% against *T. congolense* and 98.2% against *T. vivax*. The test does not show cross-reactivity with *Trypanozoon* or non-pathogenic trypanosomes [34]. Nevertheless, it is not yet known whether the test may cross-react with other antigens not yet identified or not. Farmers often use Diminazene aceturate and Isometamedium chloride upon clinical presentation, and a good number of cattle can still be positive to the test after treatment due to the relatively long half-life of circulating immunoglobulins. Trypanosome antibodies usually last 3–4 months on average after curative treatment or host self-cure, but they can last up to 13 months in some cases [33]. Another reason could be the extremely low parasitaemia, a situation in which few trypanosomes present are hidden in the blood capillaries, in the dermis and fatty tissues, but rarely occurring in the main bloodstream and still stimulate the immune system. Tests that detect antibodies are helpful in epidemiological research but not reliable for diagnostic purpose [54]. The test should be more useful for presumptive diagnosis of Trypanosomiasis [33], especially in low endemic areas.

The parasitological methods are already in use and the immunological rapid test (VerY Diag) is commercialised in the area. HRM-PCR was used as a confirmatory and most reliable method. Nevertheless, results from each diagnostic technique would purposively inform farmers and other relevant stakeholders according to their needs. Infections of *T. evansi* might be mistaken with other trypanozoon species. We recommend the use of a specific, cheaper field serological test CATT/*T.evansi* in the area, because PCR is expensive and therefore, not affordable locally.

A higher infection rate was found in Kayonza. This is probably because the district has the longest interface area with ANP compared to other districts. There were lower infections in

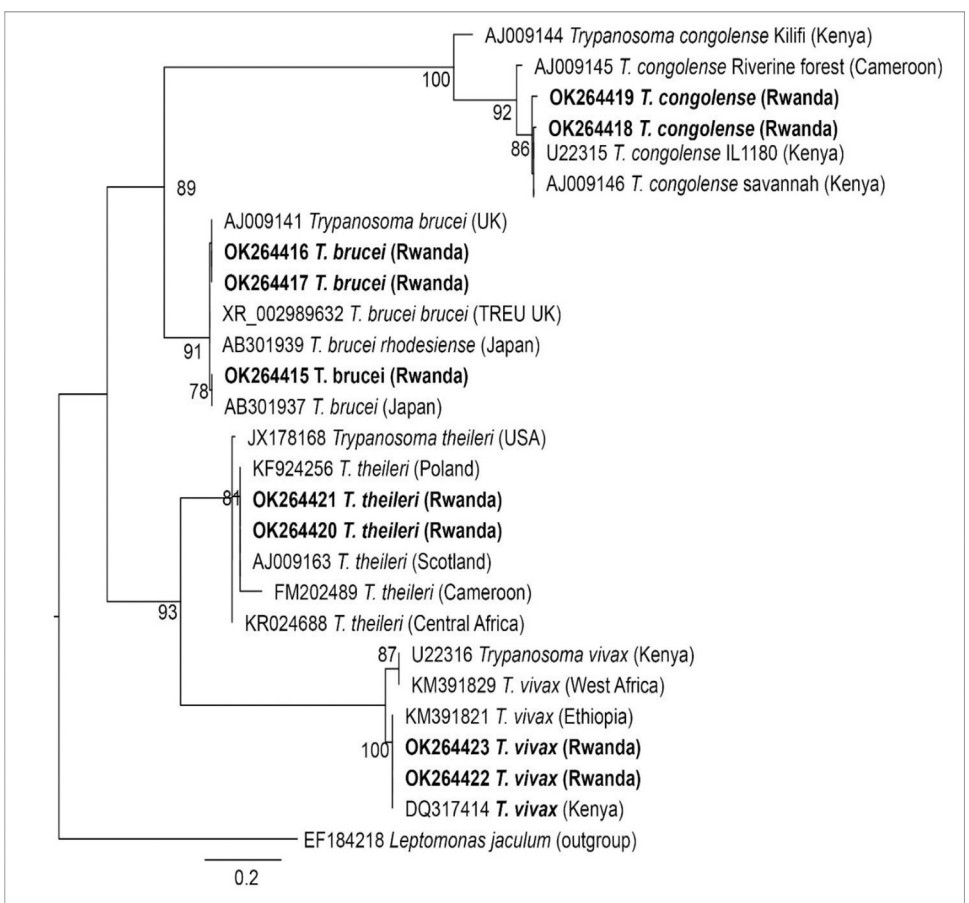

**Fig 7. Maximum likelihood phylogeny of *Trypanosoma* spp. based on partial 18S rRNA gene.** GenBank accession numbers and country of origin are indicated for each sequence. Sequences from this study are in bold. Bootstrap values at the major nodes are of percentage agreement among 1,000 replicates.The tree is rooted to outgroup sequence EF184218 (in bracket at bottom of the tree).

**Table 5. Evolutionary divergence estimates between Trypanosoma spp. of this study and the sequences and related GenBank sequences.**

| Species | Generated sequences | GenBank similarity ID (references) | p-distance |
|---|---|---|---|
| *Trypanosoma brucei* | OK264415 | AB301937—Japan | 0.000 |
| *Trypanosoma brucei* | OK264416 | AJ009141—UK | 0.000 |
| *Trypanosoma brucei* | OK264417 | XR002989632—UK | 0.000 |
| *Trypanosoma congolense* | OK264418 & OK264419 | U22315 (Savannah–IL1180)—Kenya | 0.000 |
| | | AJ009146 (Savannah)—Kenya | 0.000 |
| | | AJ009145 (Riverine forest)—Cameroon | 3.000* |
| | | AJ009144 (Kilifi)—Kenya | 17.000* |
| *Trypanosoma theileri* | OK264420 & OK264421 | AJ009163 –Scotland | 0.000 |
| | | FM202489 –Cameroon | 0.000 |
| | | KF924256—Poland | 0.000 |
| *Trypanosoma vivax* | OK264422 & OK264423 | DQ317414 (*TvL1-*Genotype)—Kenya | 0.000 |
| | | KM391821 (*TvL1-*Genotype)—Ethiopia | 0.000 |
| | | U22316 (*Tvv1-*Genotype)—Kenya | 5.000* |
| | | KM391829 –West Africa | 5.000* |

*The p-value above 1 mentioned here reflects a strain of species we got that is a mismatch with our sequences

**Table 6. Summary of VerY Diag test results.**

| VerY Diag test | NE | *T.co* | *T. vivax* | *T.co +T. vivax* | Total infections | Negative | Not specific |
|---|---|---|---|---|---|---|---|
| | 299 | 19 | 77 | 88 | 184 | 112 | 3 |
| % | | 6.4% | 26% | 29.7% | 62.1% | 37.8% | 1% |

NE = Number of animals examined; T.co = *Trypanosoma congolense*; T. vivax = *Trypanosoma vivax*; Mixed *infections counted separately from single infections*

Ankole x Friesians than Ankole cattle. Farmers tend to care more about the improved cattle than Ankole ones, hence spending more money and time on disease treatment and prevention. This could be the reason why the trypanosome infections were lower in Ankole x Friesians in the study area. However, no previous data showed different susceptibility to trypanosome infection between these types of cattle breeds in the area to align with our findings.

The PCV values were not linked to trypanosome infections. Some cattle with PCV below 26 were negative while others with the PCV values higher than 26 were positive for trypanosomes. The same observation was reported in Uganda [50]. Contrarily in endemic areas, cattle with the PCV of 26 and below are usually considered infected as a result of anaemia associated with the disease severity [55]. This could be due to low parasitaemia or simply other health conditions and malnutrition [56,57]. The infection cases found below the threshold could be associated with severe disease (high parasitaemia). Again, the constant use of trypanocides by farmers was observed in the area during the study and could prevent the disease severity, hence the absence of anaemia.

The SRA gene was not found circulating in cattle examined in this study. The SRA gene is specific for *T. b. rhodesiense* and confers resistance to survive in human serum. It serves to differentiate human infective trypanosomes and animal infective *T. b. brucei* [58]. SRA gene is expressed by trypanosomes found in both humans and animals. Therefore, cattle can serve as a reservoir for *rhodesiense* sleeping sickness. Our observations in cattle corroborate the notion that *rhodesiense* sleeping sickness may be absent from the area. The notion stems from the lack of reported cases of *rhodesiense* HAT in Rwanda for over 20 years, despite the existence of an adequate surveillance system [13,26]. Rwanda as a country meets the requirements to apply for WHO validation of HAT elimination as a public health problem at the national level [13].

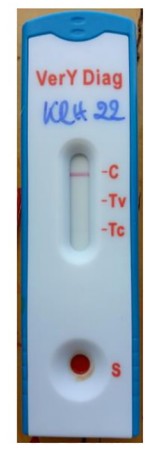 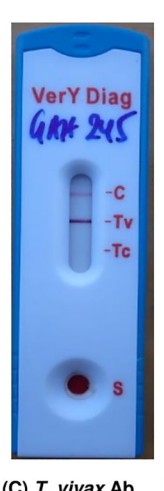 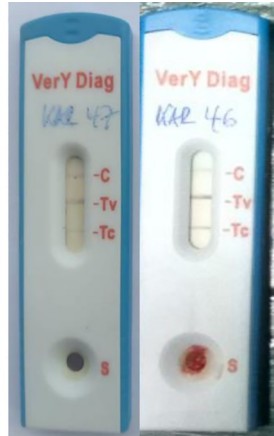

**(A) No Ab detected**   **(B)** *T. congolense* Ab    **(C)** *T. vivax* Ab    **(D) Cross reaction: Ab for** *T. congolense* **and for** *T. vivax*

**Fig 8. Rapid test cassettes used in the field.**

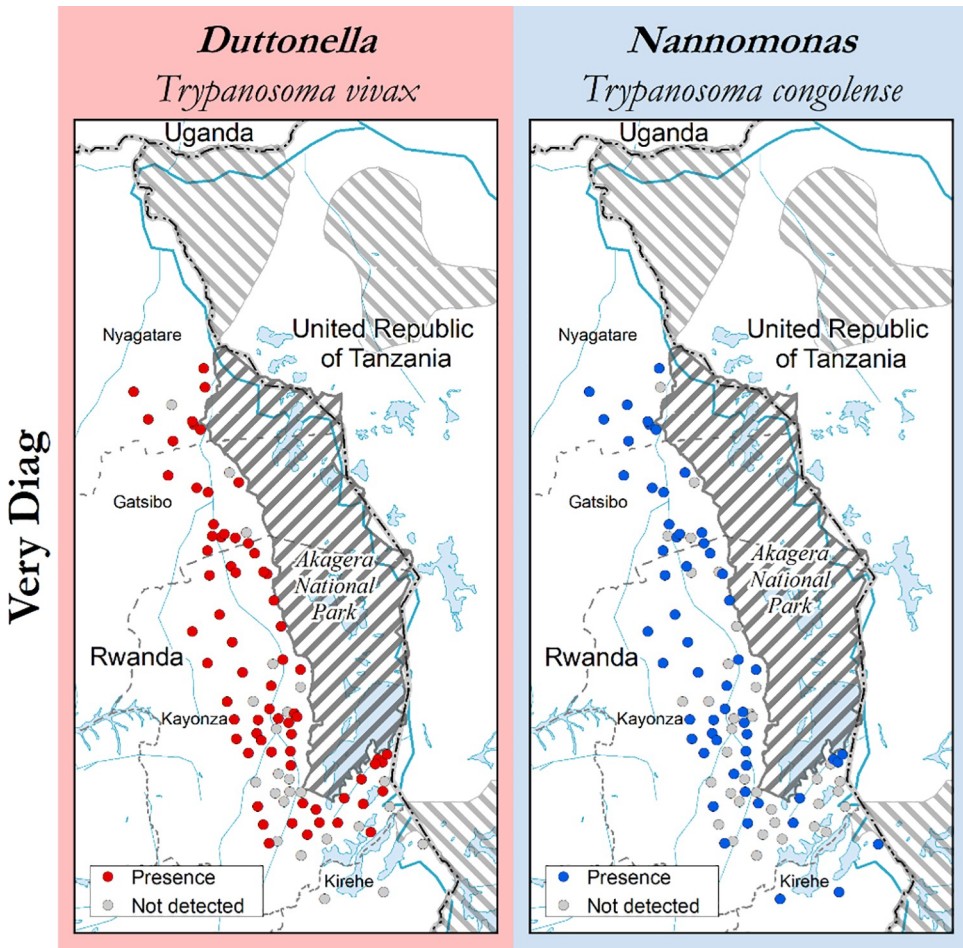

**Fig 9. Distribution of trypanosomes antibodies detected by VerY Diag.** This map was made using the data from the following GIS source files: (1) Protected Areas–WDPA https://www.protectedplanet.net/country/RWA (2) Global Administrative Unit Layers (GAUL) https://data.apps.fao.org/map/catalog/srv/eng/catalog.search;jsessionid= B7AF7A215B16770A1A67C65D97FF21CA?node=srv#/metadata/9c35ba10-5649-41c8-bdfc-eb78e9e65654 (3) Inland water bodies in Africa https://data.apps.fao.org/map/catalog/srv/eng/catalog.search;jsessionid= B7AF7A215B16770A1A67C65D97FF21CA?node=srv#/metadata/bd8def30-88fd-11da-a88f-000d939bc5d8 (4) Rivers of Africa https://data.apps.fao.org/map/catalog/srv/eng/catalog.search;jsessionid= B7AF7A215B16770A1A67C65D97FF21CA?node=srv#/metadata/b891ca64-4cd4-4efd-a7ca-b386e98d52e8.

Currently, the application process for Rwanda is in progress. However, surveillance should be maintained to confirm the absence of *rhodesiense* HAT or detect the potential re-emergence of the disease in this historically endemic area. The data presented in this study provide useful information for the validation of HAT elimination as a public health problem at the national level.

**Table 7. Sensitivity and specificity of different detection tests used.**

| Test | NE | Positives (n) | Negatives (n) | Infection rate % | Sensitivity % (95% CI) | Specificity % (95% CI) |
|---|---|---|---|---|---|---|
| Thin smear | 1037 | 61 | 976 | 5.9 | 28.9 | 99.1 |
| Buffy coat | 1037 | 88 | 949 | 8.4 | 40 | 98.6 |
| VerY Diag | 299 | 184 | 115 | 61.5 | 86 | 32.5 |
| qPCR/HRM (Reference test) | 1037 | 277 | 767 | 26.7 | NA | NA |

All positive cases are inclusive (pathogenic and non-pathogenic) N = number, NA = not applicable

This study has some limitations. First, It was designed to target mainly the risk areas around Akagera NP and a few areas around the adjacent game reserves in Tanzania. A broader study area targeting distant localities and including others along the game reserves would help to understand the impact of the distance to Akagera NP. The role of the nearby game reserves in the transmission of trypanosomes in the area would be understood as well. Although this study serves as a basis, it was only limited to one livestock species (cattle). Further investigations in other animal species and research on transmission dynamics would shed light on the full picture of trypanosomes circulating. This will contribute to a better understanding of the disease epidemiology in this setting. Another limitation was the lack of data on trypanosome infections for the rest of Rwandan territory. This affected the comparison of the current findings with the previous at a national level.

## Conclusions

The study confirms the presence of animal infective trypanosomes in the area, comprising four pathogenic (*T. congolense*, *T. vivax*, *T. brucei* and *T. evansi*) and one non-pathogenic species (*T. theileri)*. The study did not find any human-infective *T. b. rhodesiense*. The most prevalent species was *T. congolense*, which is considered the most pathogenic for cattle in sub-Saharan Africa. The PCV estimation is not always an indication of trypanosome infections. The PCV could be linked to other health conditions, not necessarily the trypanosomes.

The presence of tsetse-transmitted trypanosomes shows a continuous contact between tsetse vectors and animals. The presence of mechanically transmitted *T. evansi* and *T. theileri* indicates that disease control should not target tsetse flies only. Further investigations on the role of mechanical vectors in transmitting trypanosomes in the area are recommended. Furthermore, HAT needs to be monitored closely. The disease was once endemic in the area, cattle and wildlife constitute the potential reservoirs.

Some actions can be envisaged to promote the progressive reduction or the elimination of the AAT burden in the area [10]. An assessment of the impact of trypanosomiasis on livestock production is recommended, including a survey on the use of trypanocidal drugs. This will help farmers and policymakers rationalize control strategies and prioritise the intervention areas. Looking at the infection distribution, areas around Akagera NP as well as areas bordering the game reserves in Tanzania should be targeted for control and prevention strategies.

## Supporting information

**S1 Data set. Cattle blood samples.**
(XLSX)

## Acknowledgments

We are thankful to the administration of the districts of Kayonza, Gatsibo, and Nyagatare for the permission to conduct the fieldwork; We gratefully acknowledge the management, staff and interns of the Molecular biology and bioinformatics unit (MBBU) of icipe, for access to research facilities, collaboration, enthusiasm and good network; The Rwanda Agriculture and Animal Resources Development Board (RAB) and the School of Veterinary Medicine, the University of Rwanda for basic laboratory facilities; The Animal Production and Health Division of the Food and Agriculture Organization of the United Nations (FAO), in the framework of the Programme Against African Trypanosomiasis (PAAT), for the guidance and GIS expertise; We acknowledge the additional institutional financial support to *icipe* from the UK's Foreign, Commonwealth & Development Office (FCDO), the Swedish International

Development Cooperation Agency (Sida), the Swiss Agency for Development and Cooperation (SDC), the government of the Republic of Kenya, and the government of the Republic of Ethiopia. Last but not least, Collins Kigen (icipe) for assistance in serotyping, Eustache Musafiri, J. Claude Mpayimana, and Ernest Munyagishali; Animal Resources Officers at district and sector level, students and farmers for field facilitation, data collection and analysis.

## Disclaimer

The boundaries and names shown and the designations used on the maps presented in this paper do not imply the expression of any opinion whatsoever on the part of FAO concerning the legal status of any country, territory, city or area or of its authorities, or concerning the delimitation of its frontiers or boundaries. Dotted lines on maps represent approximate borderlines for which there may not yet be full agreement. The views expressed in this paper are those of the authors and do not necessarily reflect the views of FAO.

## Author Contributions

**Conceptualization:** Richard Gashururu S.

**Data curation:** Richard Gashururu S., Methode N. Gasana, Dennis O. Getange.

**Formal analysis:** Richard Gashururu S., Peter O. Odhiambo, Dennis O. Getange, Richard Habimana.

**Funding acquisition:** Joel L. Bargul, Daniel K. Masiga.

**Investigation:** Richard Gashururu S., Methode N. Gasana.

**Methodology:** Richard Gashururu S., Ndichu Maingi, Samuel M. Githigia, James Gashumba, Daniel K. Masiga.

**Project administration:** Joel L. Bargul, Daniel K. Masiga.

**Resources:** Joel L. Bargul, Daniel K. Masiga.

**Software:** Dennis O. Getange, Richard Habimana.

**Supervision:** Richard Gashururu S., Ndichu Maingi, Samuel M. Githigia, James Gashumba, Joel L. Bargul, Daniel K. Masiga.

**Validation:** Richard Gashururu S., Ndichu Maingi, Samuel M. Githigia, Giuliano Cecchi, Weining Zhao, James Gashumba, Joel L. Bargul, Daniel K. Masiga.

**Visualization:** Giuliano Cecchi.

**Writing – original draft:** Richard Gashururu S., Methode N. Gasana, Peter O. Odhiambo.

**Writing – review & editing:** Richard Gashururu S., Ndichu Maingi, Samuel M. Githigia, Richard Habimana, Giuliano Cecchi, Weining Zhao, James Gashumba, Joel L. Bargul, Daniel K. Masiga.

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
