## [Decision Letter · Decision Letter 0]

9 Aug 2021

Dear Mr Gashururu S.,

Thank you very much for submitting your manuscript "Occurrence, diversity and distribution of Trypanosoma infections in cattle around the Akagera National Park, Rwanda" for consideration at PLOS Neglected Tropical Diseases. As with all papers reviewed by the journal, your manuscript was reviewed by members of the editorial board and by several independent reviewers. In light of the reviews (below this email), we would like to invite the resubmission of a significantly-revised version that takes into account the reviewers' comments. 

We cannot make any decision about publication until we have seen the revised manuscript and your response to the reviewers' comments. Your revised manuscript is also likely to be sent to reviewers for further evaluation.

Sincerely,

Enock Matovu

Associate Editor

Brian Weiss

Deputy Editor

Reviewer's Responses to Questions

**Key Review Criteria Required for Acceptance?**

**Methods**

-Are the objectives of the study clearly articulated with a clear testable hypothesis stated?

-Is the study design appropriate to address the stated objectives?

-Is the population clearly described and appropriate for the hypothesis being tested?

-Is the sample size sufficient to ensure adequate power to address the hypothesis being tested?

-Were correct statistical analysis used to support conclusions?

-Are there concerns about ethical or regulatory requirements being met?

Reviewer #1: -Are the objectives of the study clearly articulated with a clear testable hypothesis stated? Partly

-Is the study design appropriate to address the stated objectives? Not well described but i have provided detailed guidelines of how this can be improved

-Is the population clearly described and appropriate for the hypothesis being tested? NO. I have offered some suggestions on how this can be done

-Is the sample size sufficient to ensure adequate power to address the hypothesis being tested? The sample calculation has not been provided. As well, the precision of the sample estimate has not been provided. I have provided guidelines of how this can be improved

-Were correct statistical analysis used to support conclusions? No. However i have provided details including ways how results should be arranged to help readers understand how the stats were calculated

-Are there concerns about ethical or regulatory requirements being met? YES

Reviewer #2: Methods are correct and well presented (no opinion about laboratory technical details which are not my strength)

Reviewer #3: Objectives of the study are clear. the sudy design and population size are adequate. Ethical and regulatory requirements have been met.

**Results**

-Does the analysis presented match the analysis plan?

-Are the results clearly and completely presented?

-Are the figures (Tables, Images) of sufficient quality for clarity?

Reviewer #1: No. However i have provided full explanation of how the current deficits can me dealt with

Reviewer #2: Results very well presented.

Reviewer #3: Results mathc the analysis plan and are clearly presented. Authors could consider to present data in function of the animal age. Presentation of some figures and tables could be improved, see detailed comments

**Conclusions**

-Are the conclusions supported by the data presented?

-Are the limitations of analysis clearly described?

-Do the authors discuss how these data can be helpful to advance our understanding of the topic under study?

-Is public health relevance addressed?

Reviewer #1: Yes; but i have also provided suggestions for improving this section

Reviewer #2: Most conclusions are supported by the study findings, with a couple exceptions that I point out below.

Some limitations are mentioned here and there, but there is not a paragraph focusing on limitations, which could be helpful as guidance for further studies.

Reviewer #3: The conclusions are supported by the data. The limitations of the study could be slightly more elaborated (such as lack of data for the rest of Rwanda). Authors discussed well how their data can be helpfull, as well as public relevance.

**Editorial and Data Presentation Modifications?**

Reviewer #1: MAJOR CHANGES. I have attached a detailed narrative of which major changes need to be effected

Reviewer #2: Abstract: 

About the PCV, authors probably mean to say no significant difference instead of no statistical significance.

The statement “should target mainly areas around Akagera NP” is not a conclusion supported by the findings, because this study only surveyed those areas.

Similarly, the statement “to advance in the Progressive Control Pathway (PCP)” is not a conclusion of this study that doesn’t deal with control strategies at all.

Methodology:

Explain why PCV was measured (e.g. to check correlations of anaemia with trypanosomes?).

Discussion:

There were less infections in crossbreeds than indigenous cattle. The authors point to a likely explanation (farmers care more about the improved cattle than indigenous ones) which makes good sense, but can they add if there are data showing different susceptibility to trypanosome infection between these types of cattle? If yes, how does it align with these findings?

It looks contradictory that after authors describe T. theileri as non-pathogenic and benign, they recommend investigations to evaluate the effect of T. theileri on cattle and /or other livestock.

When talking about surveillance, it seems correct to say that it may confirm the absence of rhodesiense HAT, but it is less clear how it can predict the potential reemergence of the disease in the area. Perhaps authors mean to say that it can detect a reemergence.

The statement “Looking at the infection distribution, Akagera NP seems to be the main source of infection” may come from other data, but it doesn’t appear to derive from this study, because the only area investigated was around Akagera NP, and the authors don’t analyse or show a difference of distance to the park of the sites with trypanosomes versus without. The maps don’t show that clearly either. And actually the contrary seems true for Tbb, found closer to Ibanda reserve than to Akagera NP.

Reviewer #3: see below

**Summary and General Comments**

Reviewer #1: i have uploaded a file of my detailed reviewers' comments. I have prefered it as an upload to keep suggested table formats

Reviewer #2: The work presented in this article is impressive and provides very valuable information for the understanding of the distribution of trypanosomal infections in cattle allowing to make inferences about trypanosome circulation beyond cattle as well.

English grammar corrections are needed throughout the manuscript. Otherwise it is very nicely presented.

The information is of high interest and it is worth publishing.

Reviewer #3: The manuscript describes the prevalence, measured by different microscopic techniques and PCR, of different animal trypanosomes in cattle, both indigenous and cross breed, around the Akagera National Park in Rwanda, which is an important livestock production area. This is an article of interest for PLOS NTD. Some minor suggestions to increase attractiveness and readability of the manuscript follow.

Abstract: 

Line 20: the sentence “The diseases occur in poor and vulnerable settings of Africa.” Is very general and can be removed. In the results section, a sentence about occurrence of trypanosomes in indigenous versus crossbreed animals might be added as this is a parameter of interest which was also studied. 

Introduction:

Line 107-109: The sentence: “Because of this epidemiological situation and the existence of an adequate surveillance system, Rwanda as a country meets the requirements to apply for WHO validation of HAT elimination as a public health problem at the national level (13). Currently, the application process for Rwanda is in progress.” Comes a bit early in the introduction and would be more appropriate in the discussion in the section about Trypanosoma brucei rhodesiense. Mention of an adequate surveillance system can be done in the previous sentence.

Material and methods

Line 135: mention the names of the indigenous and cross breed cattle 

Line 145: 145 Remove the term “warmer” in the sentence as this should be relative to something elso. The temperature range is already given a few sentences later.

Line 172: when was BCT carried out and when were specimens for PCR prepared

Line 179: how were smears stained and what was microscopic magnification used to detect trypanosomes?

Line 195: include a reference to table 1 in the section

Line 246: in the table add a column indicating for each primer species/subspecies specificity

Line 258: was VerY Diag carried out on fresh blood specimens or stored ones? Give a detailed protocol, including the volume of the blood added.

Results

Line 301: in the table it is difficult to see which column corresponds to which technique

Line 308-314: Figure 2 & 3 summarize very well the result. However the color choice for the Trypanozoon panel is not optimal, as the green spots are hardly visible. Choice of another color is indicated.

Line 315: add significance levels when comparing breeds

Line 333 figure 5: font size of the text of the graphs should be increased.

Line 351: How many animals in the non infected group had a PCV below the 26% threshold value? Add a line “non infected” in table 4. 

Results: is there anything to mention about occurrence of trypanosomes or presence of antibodies in function of animal age?

Discussion: 

Comment on trypanosome “hotspot” in Karangazi (the north West tip of Agagera?), and in particular on T evansi.

How does T evansi presence in Rwanda compare to surrounding countries? Would there be an interest to screen for T evansi antibodies for example using CATT/T evansi in Rwanda (also in the light of presence of T evansi subtype A only)?

PLOS authors have the option to publish the peer review history of their article (what does this mean?). If published, this will include your full peer review and any attached files.

Reviewer #1: Yes: Dr Muhanguzi Dennis

Reviewer #2: No

Reviewer #3: No
---

## [Editor Report · Decision Letter 1]

19 Oct 2021

Dear Mr Gashururu S.,

We are pleased to inform you that your manuscript 'Occurrence, diversity and distribution of Trypanosoma infections in cattle around the Akagera National Park, Rwanda' has been provisionally accepted for publication in PLOS Neglected Tropical Diseases.

Best regards,

Enock Matovu

Associate Editor

Brian Weiss

Deputy Editor

---

## [Editor Report · Acceptance letter]

26 Nov 2021

Dear Mr Gashururu S.,

We are delighted to inform you that your manuscript, "Occurrence, diversity and distribution of Trypanosoma infections in cattle around the Akagera National Park, Rwanda," has been formally accepted for publication in PLOS Neglected Tropical Diseases.

Best regards,

Shaden Kamhawi

co-Editor-in-Chief

Paul Brindley

co-Editor-in-Chief
